# The Role and Mechanisms of the Hypocretin System in Zebrafish (*Danio rerio*)

**DOI:** 10.3390/ijms26010256

**Published:** 2024-12-30

**Authors:** Vyacheslav Dyachuk

**Affiliations:** A.V. Zhirmunsky National Scientific Center of Marine Biology, Far Eastern Branch, Russian Academy of Sciences, 690041 Vladivostok, Russia; slavad83@gmail.com; Tel.: +7-9244205580

**Keywords:** *Danio rerio*, hypocretin, hypothalamus, sleep, neurogenesis, neurodegenerative diseases

## Abstract

Sleep is the most important physiological function of all animals studied to date. Sleep disorders include narcolepsy, which is characterized by excessive daytime sleepiness, disruption of night sleep, and muscle weakness—cataplexy. Narcolepsy is known to be caused by the degeneration of orexin-synthesizing neurons (hypocretin (HCRT) neurons or orexin neurons) in the hypothalamus. In mammals, HCRT neurons primarily regulate the sleep/wake cycle, nutrition, reward seeking, and addiction development. The hypocretin system of the brain is involved in a number of neurological disorders. The distinctive pathologies associated with the disruption of HCRT neurons are narcolepsy and cataplexy, which are caused by the loss of hypocretin neurons that produce HCRT. In Danio, the hypocretin system is also involved in the regulation of sleep and wakefulness. It is represented by a single *hcrt* gene that encodes the peptides HCRT1 and HCRT2, as well as one HCRT receptor (HCRTR), which is structurally closest to the mammalian HCRTR2. The overexpression of the *hcrt* gene in *Danio rerio* larvae causes wakefulness, whereas the physical destruction of HCRT cells or a pharmacological blockade of the type 2 hypocretin receptor leads to fragmentation of sleep in fish larvae, which is also observed in patients with narcolepsy. These data confirm the evolutionary conservatism of the hypocretin system. Thus, *Danio rerio* is an ideal model for studying the functions of HCRT neural networks and their functions.

## 1. Introduction

The HCRT (hypocretin) system is unique to vertebrates; although, in 2019, it was shown to be functional in lower chordates (lancelets) as well [1]. Since 2004, the hypocretin system has been actively studied on lower vertebrates, such as zebrafish [2,3,4,5]. Why use zebrafish for the neuroscience of the HCRT system?

In this review, we will examine the question of why danio is a suitable object for researching HCRT systems and how the data obtained can be correlated to humans. There are several reasons that zebrafish have recently emerged as a new vertebrate model of choice for neuroscientific research, having already solidified themselves in the neurodevelopment community. Zebrafish express a well-defined HCRT system that is conceptually and evolutionarily homologous with that of humans. Current studies denote the parallels between mammals and zebrafish in HCRT function. Indeed, there is an unfilled niche for evolutionary comparative studies of vertebrate behavior related to understudied neuropeptide systems, like HCRT. Zebrafish display an abundance of genetic, neurotransmitter, and hormone ligand–receptor conservation with humans.

Zebrafish *Danio rerio* is a small-sized schooling fish whose distribution range covers rivers of South Asia. This species belongs to bony fish, infraclass Teleostei, which are a monophyletic group that appeared approximately 340 million years ago [6]. Unlike other groups of vertebrates, bony fish underwent the phenomenon of genome-wide duplication [7]. Zebrafish have been a model object in developmental biology and genetics research since the late 1960s, and by the late 1980s, this animal was already widely used to elucidate the role of genes whose orthologs are important in genetic pathologies in humans [8]. Since *D. rerio* lead a diurnal life, they have become a convenient object to study the development and functions of the nervous system, such as, in particular, hypothalamic regulation of sleep [9]. Zebrafish have a rather simple and conserved brain, anatomically and functionally similar to the mammalian brain [10]. In addition, *D. rerio* suits well for the production of genetic mutants by the CRISPR/Cas9 technique [11]. Unlike other model vertebrates, transparent zebrafish embryos and larvae are very convenient for studying the nervous system, neurotracing, and investigating the fate of individual cells and even cell organelles [12,13]. Moreover, *D. rerio* is a suitable object for large-scale pharmacological, genetic, and behavioral research [14,15]. Fish sleep can be detected using behavioral criteria, such as a decrease in voluntary motor activity at night, behavioral activity reaching a peak during the day, registration of a specific posture, an increased threshold for awakening, and recovery after sleep deprivation [2,3,16,17,18,19].

## 2. Structure of the Hypocretin System in Zebrafish

Hypocretin neurons are suggested to be a quite homogeneous population of cells, but finer structural differences may also exist. There is evidence that hypocretin neurons can control numerous functions due to their genetic, morphological, and functional heterogeneity. The issue of distinguishing certain functional subpopulations of HCRT cells in the fish brain was first raised in 2021 [20]. 

The pineal gland (or *epiphysis cerebri*) in all vertebrates is involved in the regulation of diurnal and seasonal physiological changes through melatonin secretion [20,21,22]. In *D. rerio*, melatonin also mediates the pattern of circadian rhythms and stimulates sleep [19]. Projections of the hypocretin system into the pineal gland were studied on transgenic *D. rerio* lines, HCRT:EGFP, and aanat2:EGFP (arylalkylamine-N-acetyltransferase2, aanat2), which express the green fluorescent protein EGFP in HCRT cells of the ventral hypothalamus and melatonin-producing cells of the pineal gland, respectively. By crossing these fish lines, double transgenic larvae were obtained, which had axons of HCRT neurons projected into the pineal gland at 7 days of development. In adults from the offspring of double transgenic zebrafish, projections of HCRT axons into the pineal stalk and a commissural innervation of the habenula were found (Figure 1). This innervation is reported as an exceptional case of direct neuropeptidergic transmission to the fish pineal gland (Figure 2). The functionality of this projection is evidenced by the finding of hypocretin receptor mRNA in the pineal gland cells of fish larvae and adults; this indicates that the activity of pineal gland cells is modulated by HCRT [23] (Figure 1 and Figure 2).

## 3. Functions of the Hypocretin System

According to current views, hypocretin neurons stabilize wakefulness by stimulating other neural structures involved in the arousal processes in the brain; e.g., HCRT cells amplify wakefulness by stimulating norepinephrinergic neurons of the blue spot (*locus coeruleus*) [24] (Figure 2). In 2022, evidence was obtained that HCRT cells in mice can also have an effect on the stabilization of wakefulness by exerting an inhibitory effect on neurons of the ventrolateral preoptic nucleus, a brain region responsible for the initiation and maintenance of sleep [25]. The preoptic nucleus in mice contains cells that are active during sleep (Figure 2). Damage to the preoptic nucleus causes persistent insomnia, while stimulation of its cells, GABA/galanin-containing neurons, causes sleep [26,27,28,29].

The hypocretin system of *D. rerio* is very simple; it consists of 20 neurons in larvae and almost 60 neurons and adults [20]. It is mediated by the expression of a single *hcrt* gene that encodes the peptides HCRT1 and HCRT2, and also by one HCRT receptor structurally closest to the mammalian HCRTR2 [30]. As in mammals, HCRT neurons in zebrafish are localized in the lateral zones of the hypothalamus and extend projections to the telencephalon, diencephalon, mesencephalon, cerebellum, and pineal gland, as well as histamine, dopamine, melatonin, norepinephrine, serotonin, and cholinergic nuclei (Figure 2).

HCRT neurons are able to regulate the work of monoaminergic and cholinergic neurons in the hypothalamus, the brain stem, maintaining a long period of animal wakefulness. HCRT also has a strong direct excitatory effect on the cholinergic neurons of the basal part of the forebrain [31] and plays an important role in arousal. In turn, the pedunculopontine tegmental nucleus and the laterodorsal tegmental nucleus provide cholinergic communication with several brain regions and play a key role in regulating REM sleep and wakefulness phases. These regions are also strongly innervated by orexin neurons [32,33]. Electrophysiological experiments have shown that orexin A increases the frequency of excitation of cholinergic neurons [34]. Orexin neurons located in the lateral region of the hypothalamus provide a connection between the limbic system and monoaminergic or cholinergic neurons of the brain stem [35].

The overexpression of HCRT in *D. rerio* larvae causes wakefulness, whereas physical damage to HCRT cells or mutation of the receptor (HCRTR^−/−^) leads to fragmentation of sleep in fish, which is observed in patients with narcolepsy [36,37]. In general, the relative simplicity of the HCRT system makes *D. rerio* a perfect model for studying the functions of HCRT neural networks.

## 4. Effect of Genetic Ablation of HCRT Neurons

Induced genetic ablation of HCRT neurons in *D. rerio* allows for an understanding of the function of the brain hypocretin system and the cause of narcolepsy and also elucidates the relationships between cells, the role of specific cells in the embryogenesis process, and the tissue regeneration mechanisms [2]. Such studies are based on transgenic fish with the gene of bacterial nitroreductase (NTR) that catalyzes a reduction in the antibiotic metronidazole to form a cytotoxic product, leading to selective cell death. According to this method, NTR is expressed in target cells of transgenic fish by means of a tissue-specific promoter. Metronidazole added to the water with such fish causes selective death of NTR-positive cells in them [38]. Studies on the genetic ablation of HCRT neurons were conducted using a reporter stable cell line of *D. rerio*, Tg(HCRT:nfsB-EGFP), in which the HCRT promoter regulates the expressions of NTR and enhances green fluorescent protein (EGFP) simultaneously [2].

In mammals, HCRT neurons send projections onto vast regions of the brain, including dopamine, serotonin, norepinephrine, and histaminergic neurons. HCRT cells in *D. rerio* extend their projections to similar nuclei in the brain and spinal cord [5,12,16,17,39]. A study on larval whole mounts with in situ hybridization showed that in 6-day-old larvae with HCRT cell ablation, as well as in wild-type larvae, levels of tryptophan hydroxylase 1 and 2 (tph1/2, which are markers of serotonergic neurons), dopamine hydroxylase β (dβh, a marker of norepinephrinergic neurons), tyrosine hydroxylase type 1, and dopamine transporter (th1 and dat, which are markers of dopaminergic neurons) turned out to be similar. This indicates that the loss of HCRT neurons does not affect levels of mRNA marker genes and that serotonin, dopamine, and norepinephrinergic neurons remain intact [2].

## 5. Expression of the HCRT Receptor (HCRTR) in the Ablation of HCRT Neurons

HCRTR in *D. rerio* is expressed as several neural clusters in the forebrain, midbrain, and hindbrain, and also in spinal cord neurons [16,17]. An assumption was made [2] that the ablation of HCRT neurons may cause variation in the expression level of the HCRT receptor (HCRTR) gene in neurons downstream of HCRT cells and, consequently, disturbance of larvae’s behavioral activity. According to an RT-PCR analysis, the expression level of the HCRTR gene in zebrafish embryos with HCRT neuron ablation significantly increased, by 55%, compared to that in the wild type as early as 72 h post-fertilization. These data indicate that HCRTR gene expression is regulated by the innervation of HCRT neurons, and their loss causes a compensatory increase in the synthesis of HCRTR mRNA. An increase in HCRTR gene expression in target cells at a later stage of development in larvae with HCRT cell ablation can lead to changes in brain functions, which, in turn, influence behavior.

A study of daytime rhythmic activity and the threshold of awakening has not shown differences between 6-day-old larvae with HCRT cell ablation and wild-type larvae; however, the HCRT cell ablation increases the number of sleep/wakefulness transitions. During the day, larvae with HCRT neuron ablation slept 40% longer, while the number of sleep/wake transitions was 4-fold higher compared to that in the wild type [2]. These data are consistent with the narcolepsy phenomenon where intermittent night sleep and the lack of concentration during the daytime are characteristic of patients with narcolepsy [40]. The above findings show the system of HCRT neurons as conserved in fish and mammals and also suggest that the major role of these neurons consists of the regulation of changes in behavioral states, such as sleep/wake transition.

## 6. Regulation of HCRT Cell Specification in *D. rerio*

As known, mice lacking the Lhx9 transcription factor, which belongs to the LIM homeodomain protein family, have a reduced number of HCRT neurons [41]. The overexpression of Lhx9 in adult mice or neuroblastoma cell culture did not affect the number of HCRT cells and expression by HCRT cells. Therefore, the role of Lhx9 remained unclear, and the set of factors required for neurons’ differentiation into HCRT cells was unknown.

In *D. rerio*, two neuronal proteins, HCRT and pyroglutamylated RFamide Peptide (QRFP), are expressed in cells neighboring the hypothalamus and perform different functions. The neuropeptide QRFP is involved in nociception [42], feeding behavior, and motor activity [43]. Thus, the expression of HCRT and QRFP is first detected in embryos at 24 h of development in the hypothalamus where the bilateral hypothalamic nuclei consist of only 4–6 cells. Then, they grow to 10–15 cells at 120 h post-fertilization. HCRT and QRFP continue to be expressed in neighboring neurons throughout the development period but are never expressed together in the same cells. To identify the respective neurons, a *D. rerio* line was established in which HCRT cells expressed the red fluorescent protein RFP and QRFP neurons expressed the green fluorescent protein EGFP. An analysis of gene expression using microchips showed that the Lhx9 transcription factor is necessary and sufficient for the specification of neurons into HCRT cells and nearby QRFP cells. Zebrafish are known to have two sleep phases, slow interrupted sleep and rapid wave sleep, which share common features with slow and rapid sleep in mammals [44].

## 7. Heterogeneity of HCRT Neurons

A molecular marker that indicates HCRT neurons is the HCRT peptide which plays a key role in providing neuronal functions such as regulation of sleep, feeding, stress and pain responses, energy homeostasis, emotions, and reward. However, it is clear that HCRT neurons express additional key proteins that also can regulate these diverse processes. As Sagi and colleagues suggest, HCRT neurons can be divided into genetically, anatomically, and functionally different subpopulations and carry different functional loads, which makes their work more complex and diverse than we previously imagined [20]. However, it is still unclear how a supposedly homogeneous HCRT population regulates such diverse functions (sleep, wakefulness, nutrition, emotions, stress, and reward) and how changes in these patterns lead to sleep disorders and anxiety. It is possible that HCRT neurons are genetically, anatomically, and functionally heterogeneous and that subpopulations of HCRT neurons projected into different areas of the brain regulate sleep and awakening patterns and other physiological manifestations. The expression of various genetic markers in HCRT cells makes these neurons heterogeneous. A comparison between the transcriptome of the brain in patients with narcolepsy and that of a healthy human brain, as well as a comparison of mutant mice devoid of HCRT neurons with wild-type mice, showed that the IGFBP3 protein is colocalized in about 80% of HCRT neurons in mice and only 10–20% of HCRT neurons in humans [45].

## 8. Interaction Between Systems of HCRT and GnRH3 Neurons

It has been shown that in the *D. rerio* hypothalamus, HCRT neurons form connections with neurons expressing the peptide gonadoliberin (GnRH3), which indicates a close relationship between these neural systems. This was studied in 2016 on transgenic fish expressing fluorescent proteins in GnRH3 and HCRT neurons. In that study, the GnRH3 and HCRT neuronal systems, in addition to their specific neuromarkers, also co-expressed EMD (emerald green fluorescent protein) and RFP (red fluorescent protein), respectively (Figure 3). Double transgenic mutants, Tg(GnRH3:EMD) and HCRT:RFP, were obtained by crossing the stable transgenic lines Tg(GnRH3:EMD) and HCRT:RFP. In this hybrid line, the morphological interaction of two hypothalamic neural systems in the larval (8 days) and adult (3 months) stages of fish was visualized. Confocal images of the larval forebrain showed localization of GnRH3:EMD neurons in the green channel and localization of HCRT:RFP neurons in the red channel [46]. Morphologically, in zebrafish embryos, GnRH3:EMD neurons form a network of a multitude of populations, including the terminal nerve connected to the olfactory bulb, trigeminal ganglion, preoptic zone, and hypothalamus [47,48]. HCRT neurons are normally localized in only two bilaterally symmetric clusters in the hypothalamus [5]. At the larval stage of development (8 days), GnRH3:EMD neurons form two bilateral clusters that are located caudally of the aggregation of HCRT neurons but are adjacent to their zone on the lateral boundary of the HCRT neuron cluster (Figure 3). As confocal microscopy showed, both clusters of neuronal projections from GnRH3:EMD and HCRT neurons in larvae are widely distributed all over the hypothalamus with tight overlapping of neurons’ bodies and projections. The interaction between neural systems was also studied in adult fish. Confocal microscopy showed that the bodies of GnRH3:EMD are distributed in the hypothalamus and form bilateral structures of numerous processes extending from the hypothalamus’ ventral periventricular zones (Figure 3). The projections of HCRT cells are intertwined with those of GnRH3:EMD neurons, while the tight adjoining suggests direct contact between the neurons of these systems in adult fish, similar to that in larvae [48].

## 9. Expression of the HCRT Receptor in GnRH3:EMD Neurons

The distribution of the hypocretin HCRTR in the hypothalamus of adult *D. rerio* has been studied. As the results show, in wild-type fish and HCRTR^+/−^ mutants, hypothalamus cells express the receptor in similar ways, whereas homozygous HCRTR^−/−^ mutants lack the HCRTR expression. The staining of hypothalamic cells for HCRTR in adult transgenic fish of the GnRH:EMD line has shown colocalization of GnRH3:EMD neurons and HCRTR, which indicates the expression of HCRTR in GNRH3 neurons, with about 20% of GnRH3-positive cells of the periventricular region of the hypothalamus colocalized with HCRTR [46].

## 10. Diseases Associated with the Human Hypocretin System

In humans, HCRT neurons innervate several different structures (Figure 4). Axons of HCRT cells extend projections to the following centers that control wakefulness: the paraventricular nucleus, the dorsal raphe containing serotonergic neurons, the nuclei of the *tuber cinereum*, the tuberomammillary nucleus (TMN) containing histaminergic neurons, the blue spot (LC) containing norepinephrinergic neurons, and the sleep-regulating centers, such as the ventrolateral preoptic region containing γ-aminobutyric acid (GABA-ergic) neurons and the pedunculopontine tegmental nucleus (PPTg) containing cholinergic neurons. Furthermore, hypocretin neurons are connected via axons to the reward system structures, including the ventral zone of the tegmentum (VT) containing dopaminergic neurons and the raphe nucleus (RN) containing serotonergic neurons.

The hypocretin system of the brain is involved in a number of neurological disorders. The characteristic pathologies associated with HCRT neurons are narcolepsy and cataplexy, and both are caused by the loss of HCRT-producing neurons. In patients with narcolepsy, HCRT neurons die due to a highly selective neurodegenerative or autoimmune process [40,49].

Narcolepsy usually emerges in patients aged 10–20 years and begins as episodes of daytime drowsiness. In many narcolepsy patients, drowsiness is so severe that it leads to impaired concentration and wakefulness at school, at work, and also during periods of inactivity, e.g., while watching movies. Diagnosis is often made only after serious problems that arise at work, bad academic performance, or a traffic accident. It is sometimes a challenge to distinguish the daytime drowsiness caused by lack of nighttime sleep from the sleepiness in narcolepsy, but patients with narcolepsy experience daytime drowsiness even after having sufficient sleep at night. Unlike people whose daytime drowsiness is caused by poor sleep quality, e.g., in the case of obstructive sleep apnea, patients with narcolepsy feel rested after a full sleep, but drowsiness returns after 1–2 h, especially with a sedentary lifestyle. Furthermore, there is evidence that a loss of HCRT cells can also occur in neurodegenerative disorders, such as Alzheimer’s and Parkinson’s diseases, and this correlates with disease progression [50,51,52].

Disturbance of the brain hypocretin system is also found in post-traumatic injuries, dementia with Lewy bodies [53], and Prader–Willi syndrome [54,55]. In depression, insomnia can precede the disease and, thus, HCRT can be used as an early indicator [56]. The understanding of the basic neurobiological mechanisms of narcolepsy has substantially advanced after the finding that type 1 narcolepsy occurs with the death of hypocretin-producing HCRT neurons that are active during wakefulness. Target neurons, which activate hypocretin produced by HCRT cells, provide wakefulness and are located in various parts of the brain, including the cortex, basal forebrain, brainstem, and hypothalamus, and they produce histamine, dopamine, norepinephrine, and serotonin [57,58,59]. Hypocretins exert a long-term effect on target neurons, which allows maintaining alertness throughout the day. A loss of hypocretin transmission in the target regions of the brain and vice versa causes the unstable activity of their neurons, which leads to frequent falling asleep [60].

Hypocretins increase activity in the brain regions that suppress sleep and, therefore, a decrease in the signal transmission via hypocretin in narcolepsy may cause states with signs of rapid eye movement sleep, such as sleep paralysis and hypnagogic hallucinations, to develop during wakefulness [61]. Cataplexy, found in patients with narcolepsy, is characterized by a loss of muscle tone and is activated by signals that are associated with strong positive emotions, transmitted via the medial prefrontal cortex and amygdala, and it activates the pons nuclei, which causes muscle paralysis [62,63,64]. The transmission of hypocretin signals can also cause an acceleration of metabolism, sympathetic tone, and reward seeking in humans searching for narcotics. Hypocretin transmission disorder probably leads to obesity and depressive states, as is frequently observed in narcolepsy [65,66].

Takotsubo syndrome (TTS) is also suggested to be associated with hypocretin deficiency. It represents a syndrome of acute heart failure caused by emotional or physical stress. Its pathogenesis has not been fully elucidated, but the catecholamine hypothesis is the primary one to be considered among possible explanations. During the COVID-19 pandemic, an increase in the prevalence of TTS was reported in some countries. Any relationship between COVID-19 infection and TTS is unknown to date, but there is evidence indicating a possible role of hypocretin in the emergence of TTS since some symptoms of COVID-19 can be considered in the context of the dysfunction of the brain hypocretin system [67].

Schizophrenia is also associated with changes in various functions, and the regulation of the latter is mediated by hypocretin. It is suggested that the pathogenesis of schizophrenia depends on a combination of exogenous and genetic factors that are responsible for the destruction of neuronal circuits, whose effect is exerted by neurotransmitters and neuromodulators. These molecules include hypocretins. In humans, HCRT neurons are located exclusively in the hypothalamus [68] and produce two types of hypocretins, HCRT-1 and HCRT-2, with the former being the dominant hypocretin type in the brain. HCRT cells send signals to the hypothalamus, prefrontal cortex, hippocampus, dorsal raphe nucleus, and blue spot [33], acting on their neurons via two types of receptors: HCRTR-1 and HCRTR-2. These receptors are related to the G protein and are widely represented all over the central nervous system. In humans, the variability of HCRTR isomers is poorly associated with pathologies, but other disturbances of hypocretin mechanisms may have potential importance for biomedicine. For example, the impaired expression of HCRT and HCRTR, functional interactions between wild-type HCRTR and variants of the HCRT receptor, and the heterodimerization of HCRTR with opioid or cannabinoid receptors may be of interest in the context of pharmacological research [69]. The physiological processes that are affected in schizophrenia such as sleep/wake cycles, attention, cognitive functions, and energetic tonus are regulated by the hypocretin system [70]. Moreover, there are cases of false diagnoses of schizophrenia in patients with narcolepsy caused by HCRT deficiency, including those due to possible hypnagogic hallucinations in such patients [71].

As was reported, the level of HCRT-1 in blood plasma in schizophrenic patients is significantly reduced compared to that in healthy people. Postmortem studies of Hcrt-1 immunoreactivity in the hypothalamus, HCRT-1 levels in cerebrospinal fluid, and the expression of the mRNA receptor of HCRTR hypocretin in the superior frontal gyrus in patients with schizophrenia also showed some differences vs. the control group. There was a significant decrease in the amount of HCRT-1 in the hypothalamus in women with schizophrenia. Moreover, the level of mRNA HCRT2 in the superior frontal gyrus was reduced in women with schizophrenia compared to that in healthy patients. In men with schizophrenia, the expression of Rsce-K1 and Rsce-K2 mRNA had a tendency to increase compared to that in men in the control group. Thus, the central neurotransmission of hypocretins in schizophrenia decreases, especially in women, which affects its concentration in blood plasma [72].

## 11. Disadvantages of the *Danio rerio* Model in Studies of Sleep and Circadian Rhythms

*Danio rerio* are increasingly used as model organisms to study the role and mechanisms of the hypocretin system. They are small vertebrates with highly researched development, genetics, and physiology. Significantly, they have transparent embryos that allow for the visualization of cellular processes. However, using them as models in this research area has potential disadvantages. For instance, zebrafish lack certain complex hormonal and neurochemical interactions in their hypocretin systems found in higher vertebrates. So, experiments that rely on these complexities might be ineffective.

It is necessary to consider that some aspects of the hypocretin system studied in higher vertebrates are absent in the zebrafish. For example, prohormone convertase functions that generate neuropeptides from precursors are present in zebrafish but expanded in mammals. Nevertheless, translational research may complicate the interpretation of these findings. Additionally, not all neuropeptide signaling pathways exhibit similar evolutionary divergences. The zebrafish neuropeptide Y family is simpler than that of mammals, but several related behavioral and physiologic functions are retained [73]. Consequently, the absence of particular peptide isoforms or receptors raises questions about the significance of observed actions in higher vertebrates. Furthermore, it should be acknowledged that developmental genetics simplifications result in many knock-out or knock-down zebrafish lines that lack entire gene families, producing phenotypic nulls. Researchers using such models typically assume that all actions of the corresponding peptides or receptors are eliminated, but this might not be the case.

For example, zebrafish have a single HCRT homolog, while mammals have 2–3 preprohcrt isoforms that yield different peptides [74]. Zebrafish may thus lack some critically important neuropeptides that affect the interpretation of findings. In addition to these concerns, it is noted that studies providing compelling grounds to consider zebrafish valid models for investigating particular aspects of hypocretin systems also acknowledge shortcomings that need to be considered when designing experiments and interpreting the results [75].

There is functional diversity among HCRT neurons. In zebrafish and mammals, HCRT neurons have multiple physiological roles, including sleep, energy balance, and behavior. HCRT neurons are anatomically distributed differently in zebrafish compared to mammals. In mammals, HCRT neurons are found in a large cluster in the lateral hypothalamus, but in adult zebrafish, they are more scattered. This difference in distribution is likely an adaptation since other hypothalamic neuropeptides are distributed similarly in zebrafish and mammals. Zebrafish HCRT effects on sleep, locomotion, and anxiety are complex and dose dependent, at least in part because of the different anatomical distribution of HCRT receptors.

Signaling pathways activated by HCRT have been well analyzed in mammals, but less is known in zebrafish. HCRT receptors also activate adenylyl cyclase in mammals, but experiments in zebrafish have not been performed, although this may occur in zebrafish. There is generally good conservation of HCRT signaling pathways, but there are differences in the details of neuropeptide signaling that need to be worked out. Overall, zebrafish will be a useful model system for studying the evolution of the function of HCRT because of the different anatomical distribution and receptor types. Zebrafish also provide insights into desired sleep and metabolic disorders because HCRT neurons are associated with functional heterogeneity. HCRT neurons are evolutionarily conserved, but their function has diversified. Functional heterogeneity may have general implications for understanding how similar neural circuits cope with a wide variety of states, such as sleep and metabolic disorders.

## 12. Conclusions

Danio is a convenient model for studying the hypocretin system and has the potential to improve the study of neurobiology and human health. Hypocretin is involved in many physiological processes and is best known for its role in regulating sleep and behavior. *Danio rerio* fish, a relatively new but successful model organism, have the potential to accelerate research on the hypocretin system due to the simplicity of conducting experiments based on their reliable genetics, rapid embryonic development, genetic controllability, and the possibility of screening mutations. In recent years, knowledge about the anatomical distribution of HCRT and their receptors and functions in zebrafish has expanded significantly, including knowledge about several levels of functions at the systemic level. Ultimately, research aimed at deepening our understanding of the HCRT system in Danio fish may also open up new data that may improve human health.

## Figures and Tables

**Figure 1 ijms-26-00256-f001:**
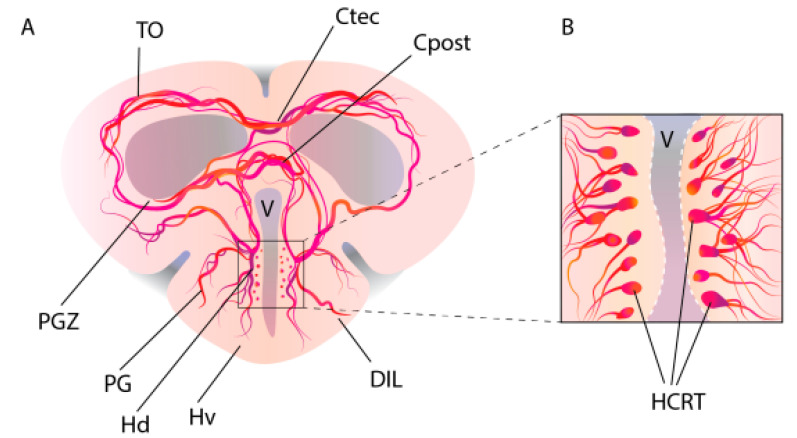
Diagram of a frontal section of an adult *Danio rerio* brain at the level of the hypothalamus. (**A**) Localization of HCRT neurons and projection of their axons into various brain structures. (**B**) HCRT neurons form two symmetrical clusters in the periventricular zone of the hypothalamus. Letter designations are as follows: Cpost, *commissura posterior*; Ctec, *commissura tecti*; DIL, diffuse nucleus of the hypothalamus; Hd, hypothalamus dorsal zone; Hv, hypothalamus ventral zone; PG, preglomerular nucleus of the hypothalamus; PGZ, periventricular gray zone of *tectum opticum*; TO, *tectum opticum*; V, ventriculus.

**Figure 2 ijms-26-00256-f002:**
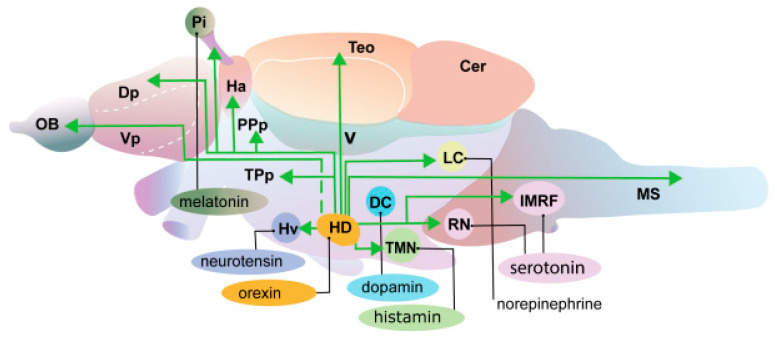
Diagram of the neural projections of the hypocretin HCRT neurons in *Danio rerio*. Letter designations are as follows: DC, diencephalic cluster of dopamine neurons; Dp, dorsal part of telencephalon; Ha, habenula; Pi, pineal gland; HD, hypothalamus dorsal zone; Hv, ventral hypothalamus; IMRF, intermediate reticular formation; LC, blue spot; OB, olfactory bulb; PPp, parvocellular preoptic nucleus; TMN, tuberomammillary nucleus; TeO, *tectum opticum*; TPp, periventricular nucleus of *tuberculum posterior*; RN, raphe nuclei; V, ventricle; Vp, ventral part of the telencephalon.

**Figure 3 ijms-26-00256-f003:**
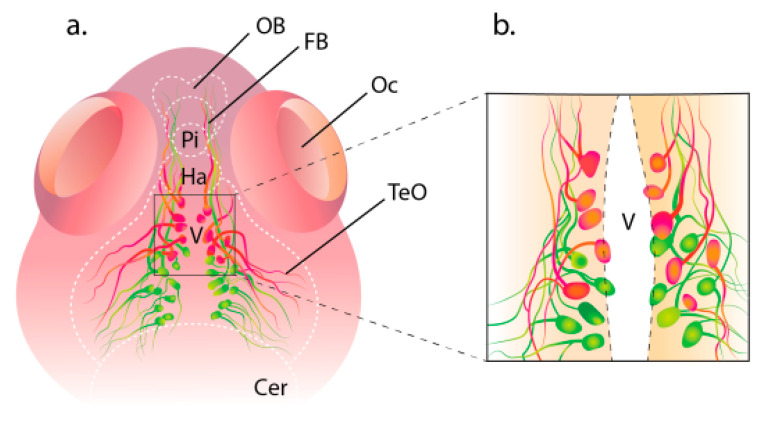
Diagram of the localization of HCRT neurons (in red) and GnRH3 (in green) in *Danio rerio*. (**a**) Projections of HCRT and GnRH3 systems in *D. rerio* larva brain. (**b**) Periventricular zone of dorsal hypothalamus close up. Letter designations are as follows: Cer, cerebellum; FB, forebrain; OB, olfactory bulb; TeO, *tectum opticum*; V, ventricle.

**Figure 4 ijms-26-00256-f004:**
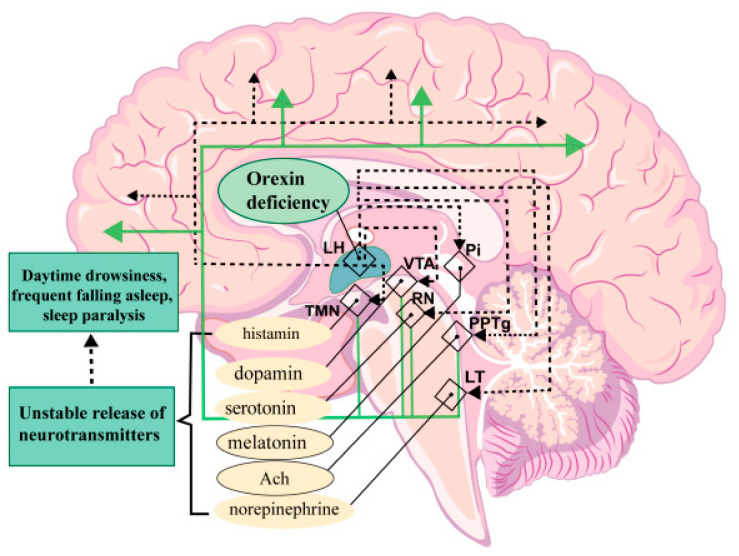
Mechanism of narcolepsy progression. Neurons of the lateral hypothalamus (LH), normally producing hypocretin, help maintain the wakefulness state by stimulating other neurons that contribute to maintaining sleep and wakefulness in brain structures such as the pineal gland (Pi), ventral tegmental area (VT), tuberomammillary nuclei (TMN), raphe nuclei (RN), pedunculopontine tegmental nucleus (PPTg), and blue spot (*locus coeruleus*, LC). Loss of hypocretin neurons is accompanied by a disturbance of neurons’ activity in the above-listed/these/certain areas, which can lead to drowsiness, frequent falling asleep, and sleep paralysis.

## Data Availability

All the data used for the analyses in this report are available from the corresponding author upon reasonable request.

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
