# Peer review of "The Role and Mechanisms of the Hypocretin System in Zebrafish (Danio rerio)"

_ijms, 2024, doi:10.3390/ijms26010256_

Round 1
Reviewer 1 Report
Comments and Suggestions for Authors
Manuscript title:
Orexin System of Zebrafish, Danio rerio
Dyachuk V.A.
This review article is an interesting topic, and the author explains that zebrafish is a model for studying the hypocretin (orexin) system and its physiological roles.
The text is confused with orexin or hypocretin words. The author must stick with one name.
The figure legends should have one pattern of abbreviation – alphabetical order. Now, it is irregularly arranged.
The references do not follow the journal criteria.
In the second paragraph of page 2, there are mistakes with the PDF conversion.
Page 6, lines 216 and 217. Hypocretin pathology in human narcolepsy. The references are Peyron et al. 2000 and Thannickal et al. 2000. Sakurai and Scammell to be removed.
The article ended without a conclusion.
Author Response
Rev 1
Dyachuk V.A.
This review article is an interesting topic, and the author explains that zebrafish is a model for studying the hypocretin (orexin) system and its physiological roles.
The text is confused with orexin or hypocretin words. The author must stick with one name.
Reply:
Thank you for your comment. We have brought everything to uniformity as hypocretin/
The figure legends should have one pattern of abbreviation – alphabetical order. Now, it is irregularly arranged.
Reply:
Done
The references do not follow the journal criteria.
Reply:
Changed
In the second paragraph of page 2, there are mistakes with the PDF conversion.
Reply:
Sorry for this technical error. Fixed
Page 6, lines 216 and 217. Hypocretin pathology in human narcolepsy. The references are Peyron et al. 2000 and Thannickal et al. 2000. Sakurai and Scammell to be removed.
Reply:
Deleted
The article ended without a conclusion.
Reply:
I have added the conclusion in the end of MS.
Reviewer 2 Report
Comments and Suggestions for Authors
The manuscript "Orexin System of Zebrafish, Danio rerio“ explores on significance of Danio rerio as a model to study the functions of hypocretin neural networks and their functions. It has been frequently reported that the orexin/hypocretin (ORX) system is involved in various physiological processes as well as in the control of sleep and wakefulness. Studies have speculated that ORX system may drive the aminergic and cholinergic activities that control sleep and wakefulness states. However, the biological mechanisms and relevance of the interactions between the neurotransmitter systems are poorly understood. The submitted manuscript, analyzing the available literatures, explores these aspects to some extent.
After going through the manuscript, I have following comments for the authors.
1. I would suggest the authors to elaborately discuss the cholinergic and aminergic input to the ORX system.
2. The manuscript needs to be structured: The introduction is bulky, the objective of the study is mentioned neither in the manuscript nor in the text. The abstract also needs restructuring, it is composed so that the interpreted results are not from external studies but from own experiments. Conclusion is also missing.
3. The title of the manuscript does not reflect the spectrum of the study. It should be something meaningful (eg. The Role and Mechanisms of the Orexin System in Zebrafish (Danio rerio) or Exploring the Orexin System in Zebrafish (Danio rerio): Functional Insights and Implications, …….)
Comments on the Quality of English Language
Improvement of the language including grammatical corrections and syntax adjustments suggested.
Author Response
Rev 2
The manuscript "Orexin System of Zebrafish, Danio rerio“ explores on significance of Danio rerio as a model to study the functions of hypocretin neural networks and their functions. It has been frequently reported that the orexin/hypocretin (ORX) system is involved in various physiological processes as well as in the control of sleep and wakefulness. Studies have speculated that ORX system may drive the aminergic and cholinergic activities that control sleep and wakefulness states. However, the biological mechanisms and relevance of the interactions between the neurotransmitter systems are poorly understood. The submitted manuscript, analyzing the available literatures, explores these aspects to some extent.
After going through the manuscript, I have following comments for the authors.
- I would suggest the authors to elaborately discuss the cholinergic and aminergic input to the ORX system.
Reply
I added a paragraph about the role the cholinergic and aminergic input to the ORX system
- The manuscript needs to be structured: The introduction is bulky, the objective of the study is mentioned neither in the manuscript nor in the text. The abstract also needs restructuring, it is composed so that the interpreted results are not from external studies but from own experiments. Conclusion is also missing.
Reply
We have performed textual changes
- The title of the manuscript does not reflect the spectrum of the study. It should be something meaningful (eg. The Role and Mechanisms of the Orexin System in Zebrafish (Danio rerio) or Exploring the Orexin System in Zebrafish (Danio rerio): Functional Insights and Implications, …….)
Reply
Changed
Comments on the Quality of English Language
Improvement of the language including grammatical corrections and syntax adjustments suggested.
Reply
Verified by a native speaker
Reviewer 3 Report
Comments and Suggestions for Authors
The manuscript "Orexin System of Zebrafish, Danio rerio" is not structured, and its format does not permit evaluation. Page 2 is impossible to read. Please revise the manuscript to be evaluated.
Comments on the Quality of English LanguageThe English is good.
Author Response
Rev3
Comments and Suggestions for Authors
The manuscript "Orexin System of Zebrafish, Danio rerio" is not structured, and its format does not permit evaluation. Page 2 is impossible to read. Please revise the manuscript to be evaluated.
Comments on the Quality of English Language
The English is good.
Reply
Thanks for the comments. We have corrected page 2 and made the appropriate textual changes
Reviewer 4 Report
Comments and Suggestions for Authors
The manuscript entitled “Orexin System of Zebrafish, Danio rerio” discusses the insights of the overexpression of the hcrt gene in Danio larvae causes wakefulness, whereas physical destruction of HCRT cells or pharmacological blockade of the type 2 orexin receptor leads to fragmentation of sleep in fish larvae, which is also observed in patients with narcolepsy. The presented data confirm the conservatism of the orexin system. The review is well-written and I recommend for publication after minor revision.
The below revisions are recommended:
- The title does not sound meaningful. It does not reflect the materials that have been discussed in the manuscript. I recommend to revise the title meaningfully.
- There are many reports in the literature that include similar research objectives. Cite some recent reports in the manuscript.
- The manuscript must be thoroughly checked, and the quality of the language must be improved. There are numerous grammatical mistakes.
- Uniformity (font and size) should be mentioned throughout the manuscript, including the schemes and figures. The authors are encouraged to check the journal IFA.
Author Response
Rev4
The manuscript entitled “Orexin System of Zebrafish, Danio rerio” discusses the insights of the overexpression of the hcrt gene in Danio larvae causes wakefulness, whereas physical destruction of HCRT cells or pharmacological blockade of the type 2 orexin receptor leads to fragmentation of sleep in fish larvae, which is also observed in patients with narcolepsy. The presented data confirm the conservatism of the orexin system. The review is well-written and I recommend for publication after minor revision.
The below revisions are recommended:
- The title does not sound meaningful. It does not reflect the materials that have been discussed in the manuscript. I recommend to revise the title meaningfully.
Reply
We have changed the title of the MS “The Role and Mechanisms of the Hypocretin System in Zebrafish (Danio rerio)
- There are many reports in the literature that include similar research objectives. Cite some recent reports in the manuscript.
Reply
Done
- The manuscript must be thoroughly checked, and the quality of the language must be improved. There are numerous grammatical mistakes.
Reply
Verified by a native speaker
- Uniformity (font and size) should be mentioned throughout the manuscript, including the schemes and figures. The authors are encouraged to check the journal IFA
Reply
Done
Reviewer 5 Report
Comments and Suggestions for Authors
I my opinion this MS is very interesting and important because it describes a model fish, and the research can be translated to mammals, including humans. The study concerns the role of the orexin system (HCRT) in the regulation of sleep and wakefulness and the use of the fish Danio rerio as a research model for the analysis of the function of orexin neurons, the disruption of which leads to diseases such as narcolepsy and cataplexy. The author indicated that Danio rerio, due to the similarities in the structure of the HCRT system to mammals, is an excellent model for the analysis of sleep mechanisms and disorders related to it.
I believe that these studies are of great practical importance, because understanding the function and disorders of the orexin system in a model such as Danio rerio. These studies may contribute to the development of new therapies for the treatment of narcolepsy and other sleep disorders, also in humans. However, the MS contains some editing errors, which I have noted in the text.

Author Response
Rev5
I my opinion this MS is very interesting and important because it describes a model fish, and the research can be translated to mammals, including humans. The study concerns the role of the orexin system (HCRT) in the regulation of sleep and wakefulness and the use of the fish Danio rerio as a research model for the analysis of the function of orexin neurons, the disruption of which leads to diseases such as narcolepsy and cataplexy. The author indicated that Danio rerio, due to the similarities in the structure of the HCRT system to mammals, is an excellent model for the analysis of sleep mechanisms and disorders related to it.
I believe that these studies are of great practical importance, because understanding the function and disorders of the orexin system in a model such as Danio rerio. These studies may contribute to the development of new therapies for the treatment of narcolepsy and other sleep disorders, also in humans. However, the MS contains some editing errors, which I have noted in the text.
Reply
Thank you so much for the comments and edits to the text. Thank you for your time spent on the text. This is very valuable and important for us. Thank you so much. Your edits are all taken into account
Round 2
Reviewer 1 Report
Comments and Suggestions for Authors
ijms-3328162
Title: The role and mechanisms of the hypocretin system in Zebrafish (D.rerio)
Author: Dyachuh V.A.
The revised manuscript changed according to suggestions.
There are a few things to consider :
The abbreviation used for hypocretin is not uniform in the text
HCRT, Hcrt. It should be uniform in the text.
Line 99: HcrtR receptor
Receptor is repetition
Line 166: 3-neurons
It is GnRH3 -neurons
Line 201: HcrtR receptor
Receptor is repetition
Line 305: HCR-1 has to correct
Line 310: HRT has to correct
Line 318: Riske -K2 has to correct
Reference 22 is not in the text.
Comments on the Quality of English LanguageThe quality of the language is to be improved.
Author Response
Reviewer 1
The revised manuscript changed according to suggestions.
There are a few things to consider :
The abbreviation used for hypocretin is not uniform in the text
HCRT, Hcrt. It should be uniform in the text.
Line 99: HcrtR receptor
Receptor is repetition
Line 166: 3-neurons
It is GnRH3 -neurons
Line 201: HcrtR receptor
Receptor is repetition
Line 305: HCR-1 has to correct
Line 310: HRT has to correct
Line 318: Riske -K2 has to correct
Reference 22 is not in the text.
Comments on the Quality of English Language
Te quality of the language is to be improved.
Reply
Dear reviewer!Thank you so much for such a thorough reading and detection of typos and errors in the text. All your comments have been taken into account and corrected by me.
1)The contractions of receptors and ligands are brought to uniformity
2) Added a link
3) fixed molecules
4)fixed typos in the text
5) English is verified by a native speaker
Thank you for your time and for checking the text so thoroughly.
Author
Reviewer 2 Report
Comments and Suggestions for Authors
Thank you for doing necessary corrections. I am happy with this version.
Author Response
Reviewer 2
Thank you for doing necessary corrections. I am happy with this version.
Reply
Thank you so much
Reviewer 3 Report
Comments and Suggestions for Authors
The manuscript "The Role and Mechanisms of the Hypocretin System in
Zebrafish (Danio rerio)" is well written.
1. The research examines the advantages of utilizing zebrafish while neglecting to consider potential disadvantages, like the lack of certain human-like intricacies in their hypocretin systems.
2. Please enhance figures with clearer annotations and align them more closely with the textual content.
3. To clarify, expand the discussion on the functional heterogeneity of HCRT neurons with a more in-depth analysis of existing literature.
I recommend a minor revision.
Author Response
Reviewer 3
The manuscript "The Role and Mechanisms of the Hypocretin System in
Zebrafish (Danio rerio)" is well written.
- The research examines the advantages of utilizing zebrafish while neglecting to consider potential disadvantages, like the lack of certain human-like intricacies in their hypocretin systems.
- Please enhance figures with clearer annotations and align them more closely with the textual content.
- To clarify, expand the discussion on the functional heterogeneity of HCRT neurons with a more in-depth analysis of existing literature.
I recommend a minor revision.
Reply
Dear reviewer. Thank you for your comments.
We added more information and drew attention to the shortcomings of the zebrafish model and the heterogeneity of neurons in short, because this can be a discussion and the subject of a separate article.
Reviewer 5 Report
Comments and Suggestions for Authors
In my opinion MS is very interesting and can be published in its current form.
Author Response
Reviewer 5
In my opinion MS is very interesting and can be published in its current form.
Reply
Thank you so much